# Structural and Magnetic Properties of Co_0.5_Ni_0.5_Ga_0.01_Gd_0.01_Fe_1.98_O_4_/ZnFe_2_O_4_ Spinel Ferrite Nanocomposites: Comparative Study between Sol-Gel and Pulsed Laser Ablation in Liquid Approaches

**DOI:** 10.3390/nano11092461

**Published:** 2021-09-21

**Authors:** Munirah A. Almessiere, Sadik Güner, Yassine Slimani, Mohammed Hassan, Abdulhadi Baykal, Mohammed Ashraf Gondal, Umair Baig, Sergei V. Trukhanov, Alex V. Trukhanov

**Affiliations:** 1Department of Physics, College of Science Imam Abdulrahman Bin Faisal University, P.O. Box 1982, Dammam 31441, Saudi Arabia; 2Department of Biophysics, Institute for Research and Medical Consultation (IRMC), Imam Abdulrahman Bin Faisal University, P.O. Box 1982, Dammam 31441, Saudi Arabia; 3Institute of Inorganic Chemistry, RWTH Aachen University, 52074 Aachen, Germany; s.guner@ac.rwth-aachen.de; 4Laser Research Group, Physics Department & Center of Excellence in Nanotechnology, King Fahd University of Petroleum & Minerals (KFUPM), Dhahran 31261, Saudi Arabia; scholarhassan@gmail.com; 5Department of Nanomedicine Research, Institute for Research and Medical Consultation (IRMC), Imam Abdulrahman Bin Faisal University, P.O. Box 1982, Dammam 31441, Saudi Arabia; abaykal@iau.edu.sa; 6Laser Research Group, Physics Department & IRC-Hydrogen and Energy Storage, King Fahd University of Petroleum and Minerals, Dhahran 31261, Saudi Arabia; magondal@kfupm.edu.sa; 7K.A. CARE Energy Research and Innovation Center, King Fahd University of Petroleum and Minerals, Dhahran 31261, Saudi Arabia; 8Interdisciplinary Research Center for Membranes and Water Security, King Fahd University of Petroleum and Minerals, Dhahran 31261, Saudi Arabia; umairbaig@kfupm.edu.sa; 9Laboratory of Magnetic Films Physics, Scientific-Practical Materials Research Centre of National Academy of Sciences of Belarus, 220072 Minsk, Belarus; sv_truhanov@mail.ru; 10Department of Electronic Materials Technology, National University of Science and Technology MISiS, 119049 Moscow, Russia; 11L.N. Gumilyov Eurasian National University, Nur-Sultan 010008, Kazakhstan

**Keywords:** spinel ferrites, magnetic nanocomposite, sol-gel method, pulsed laser ablation in liquid

## Abstract

In this study, the samples of the ZnFe_2_O_4_ (ZFO) spinel ferrites nanoparticles (SFNPs), Co_0.5_Ni_0.5_Ga_0.01_Gd_0.01_Fe_1.98_O_4_ (CNGaGdFO) SFNPs and (Co_0.5_Ni_0.5_Ga_0.01_Gd_0.01_Fe_1.98_O_4_)*_x_*/(ZnFe_2_O_4_)*_y_* (*x*:*y* = 1:1, 1:2, 1:3, 2:1, 3:1 and 4:1) (CNGaGdFO)*_x_*/(ZFO)*_y_* spinel ferrite nanocomposites (NC) have been synthesized by both sol-gel and Green pulsed laser ablation in liquid (PLAL) approaches. All products were characterized by X-ray powder diffraction (XRD), scanning and transmission electron microscopies (SEM and TEM), elemental mappings and energy dispersive X-ray spectroscopy (EDX). It was objected to tune the magnetic properties of a soft spinel ferrite material with a softer one by mixing them with different fractions. Some key findings are as follows. M-H investigations revealed the exhibition of ferrimagnetic phases for all synthesized samples (except ZnFe_2_O_4_) that were synthesized by sol-gel or PLAL methods at both 300 K and 10 K. ZnFe_2_O_4_ ferrite NPs exhibits almost paramagnetic feature at 300 K and glass-like phase at very low temperatures below 19.23 K. At RT analyses, maximum saturation magnetization (*M*_S_) of 66.53 emu/g belongs to nanocomposite samples that was synthesized by sol-gel method and *x:y* ratio of 1:3. At 10 K analyses, *M*_S_,_max_ = 118.71 emu/g belongs to same nanocomposite samples with ratio of 1:3. Maximum coercivities are 625 Oe belonging to CNGaGdFO and 3564 Oe belonging to NC sample that was obtained by sol-gel route having the 3:1 ratio. Squareness ratio (SQRs = *M*_r_/*M*_S_) of NC sample (sol-gel, 4:1 ratio) is 0.371 as maximum and other samples have much lower values until a minimum of 0.121 (laser, 3:1) assign the multi-domain wall structure for all samples at 300 K. At 10 K data, just CNGaGdFO has 0.495 SQR value assigning single domain nature. The maximum values of effective crystal anisotropy constant (*K*_eff_) are 5.92 × 10^4^ Erg/g and 2.4 × 10^5^ Erg/g belonging to CNGaGdFO at 300 K and 10 K, respectively. Further, this sample has an internal anisotropy field *H*_a_ of 1953 Oe as largest at 300 K. At 10 K another sample (sol-gel, 3:1 ratio) has H_a,max_ of 11138 Oe which can also be classified as a soft magnetic material similar to other samples. Briefly, most magnetic parameters of NCs that were synthesized by sol-gel route are stronger than magnetic parameters of the NCs that were synthesized by PLAL at both temperatures. Some NC samples were observed to have stronger magnetic data as compared to magnetic parameters of Co_0.5_Ni_0.5_Ga_0.01_Gd_0.01_Fe_1.98_O_4_ NPs at 10 K.

## 1. Introduction

Magnetic nano particles especially nano-sized spinal ferrites (NSFs) are promising magnetic materials due to their extensive applications in medicine, photocatalysis, microwave absorption and energy storage [1,2,3]. The NSFs are complex metal oxides with structural formula (A)[B_2_]O_4_ that can be portrayed as a cubic closed pack of oxygen ions. The round brackets (A) represent tetrahedral interstitial sites and square brackets [B] correspond to larger octahedral sites. Both of these sites are occupied by cations with divalent ions occupying tetrahedral and trivalent ions occupying octahedral sites [4]. The magnetic moments of the cations occupying octahedral lattice sites are oriented in the same direction. Whereas the magnetic moment of the cations at tetrahedral sites are oriented anti parallel to that of the cations at octahedral sites. The resultant of these two magnetic moments give rise to the net magnetization in spinel ferrites [5].

In the bulk form, ZnFe_2_O_4_ has a normal spinel structure with Zn^2+^ ions occupying tetrahedral lattice sites and Fe^3+^ ions at the octahedral sites, indicating the form (Zn^2+^)[Fe^3+^_2_]O_4_. However, it has been reported that different synthesis methods for ferrite nanoparticles can alter the structural parameters, composition, cationic distribution and magnetic properties of the nano ferrites that cannot be observed in their bulk forms [6]. The electric, magnetic and optical properties of the NSF are associated with the distribution and occupancy of the cations in the crystal structure. Such orientations take place at the atomic level depending upon the reaction conditions and synthesis method [7]. The desire to control the physiochemical properties of NSF during the synthesis process remains a tricky task.

Spinel ferrites that consist of a combination of two or more divalent metal ions are referred to as mixed ferrites. The surface properties of the mixed ferrites are influenced by the distribution of the cations between tetrahedral and octahedral sites [8]. In addition to that, further composition of two different metal ferrites with hard and soft magnetic nature are expected to tune the magnetic properties of the composite in a unique way along with improving the other properties [9]. Hussain et al. reported the substitution of Co and Gd to ZnFe_2_O_4_ using co-precipitation method [10]. The addition of Co and Gd not only enhanced the saturation magnetization but also increased the coercivity. Heiba et al. have reported the synthesis of Ga substituted Ni ferrite using citrate method [11]. The substitution of Ga atoms to NiFe_2_O_4_ reduced the crystallite size and increased the dielectric nature of the Ni ferrite. Hamdy et al. reports the synthesis of Zn, Gd and Ga doped Nickel ferrite using two different synthesis routes as solid-state reaction and citrate method [12]. The samples synthesized using solid state reaction showed magnetic order whereas the samples prepared through citrate method exhibited super magnetic nature. These results show that the structural and magnetic properties of the nano ferrites depend mainly on the reactants, synthesis methods, initial concentration and thermal treatment.

The laser ablation in liquids has demonstrated the following advantages: (i) a chemically “simple and clean” synthesis, (ii) an ambient conditions not extreme temperature and pressure, and (iii) the new phase formation of nanocrystals may involve in both liquid and solid, (iv) usage of surfactants is not necessary to produce dispersed NPs, in contrast with chemical methods of nanoparticle synthesis, (v) simultaneous preparation of nanoparticle and NSs can be performed in a single experiment within in a few minutes [13]. These advantages allow us to combine selected solid targets and liquid to fabricate compound nanostructures with desired functions [14]. Additionally, laser ablation in liquids over other techniques for the nanocrystal preparation with the metastable phases are a chemically “simple and clean” preparation (reduced by product formation, simpler starting materials, no need for catalyst, etc.), an ambient condition and a variety of metastable phases that may not be attainable by the same mild preparation methods. Therefore, PLAL is a simple platform to produce not only the NPs of metals, but also the NPs of semiconductors, alloys, oxides, magnetic materials, biaxial hetero structures and core-shell type, etc. [15].

In this study, the effect of synthesis method (sol-gel auto-combustion and Green pulsed laser ablation in liquid (PLAL) methods) on the structural and magnetic properties (room and low temperature magnetization and Temperature dependence of ZFC and FC magnetizations) of the hard/soft mixed ferrite nanocomposites (Co_0.5_Ni_0.5_Ga_0.01_Gd_0.01_Fe_1.98_O_4_)*_x_*/(ZnFe_2_O_4_)*_y_* (*x:y* = 1:1, 1:2, 1:3, 2:1, 3:1 and 4:1) NCs) which were synthesized through one-pot sol-gel auto-combustion and Green pulsed laser ablation in liquid (PLAL) methods. The structure, morphology and composition of the final composite material is studied through XRD, SEM, EDS and HR-TEM.

## 2. Materials and Methods

### 2.1. Synthesis and Characterization

The samples of the ZFO NPs, CNGaGdFO NPs individually, and (CNGaGdFO)*_x_*/(ZFO)*_y_* (*x:y* = 1:1, 1:2, 1:3, 2:1, 3:1 and 4:1) NCs were produced by two ways, first was one-pot sol–gel auto-combustion and second was Green pulsed laser ablation in liquid (PLAL) method.

#### 2.1.1. One-Pot Sol–Gel Auto-Combustion Method

A precise ratios of metal nitrates Zn(NO_3_)_2_ and Fe(NO_3_)_3_·9H_2_O were mixed with 50 mL DI water under continuous stirring at 70 °C for 35 min to prepare the ZFO solution. To synthesize the CNGaGdFO, Co(NO_3_)_2_, Ni(NO_3_)_2_, Ga(NO_3_)_3_ and Gd(NO_3_)_3_, NH_3_ and Fe(NO_3_)_3_·9H_2_O were mingled with 50 mL DI water under continuous stirring at 70 °C for 35 min. Both ZFO and CNGaGdFO solutions were added to each other in different fractions to make (CNGaGdFO)*_x_*/(ZFO)*_y_* (*x:y* = 1:1, 1:2, 1:3, 2:1, 3:1 and 4:1) NCs and mixed along with high purity citric acid with stirring at 70 °C for 35 min. A neutral pH at 7 was achieved through ammonia solution then increasing the temperature up to 180 °C for 50 min and at 380 °C until obtain black color powder. The procurers were calcining at 800 °C for 5 h.

#### 2.1.2. Green Pulsed Laser Ablation in Liquid (PLAL) Method

Firstly, the ZFO and CNGaGdFO NPs were fabricated separately by sol-gel auto-combustion technique as depicted in the beginning. To synthesis of (CNGaGdFO)*_x_*/(ZFO)*_y_* (*x:y* = 1:1, 1:2, 1:3, 2:1, 3:1 and 4:1) NCs by PLAL a precise ratio of ZFO and CNGaGdFO NPs in two beakers with 25 mL of DI water and subjected them to ultrasonication for 20 min, and afterwards, these solutions were mixed in a one beaker, and using ultrasonic vibration one more time for 1 h. The NC mixtures was exposed to high intense pulsed laser irradiation from Q-Switched Nd-YAG laser (Quantel-Brilliant B) (pulse duration 9 ns, 10 Hz repetition rate, 532 nm wavelength and 300 mJ pulse energy) using a routing optics and focusing lens for about 40 min, while the sample under stirring. The NCs resulted after the laser irradiation were dried for 2 h at 120 °C. The NCs with different mass ratios were produced by the same method.

The structure of NCs with different ratios synthesized by One-pot sol-gel auto-combustion and PLAL methods was verified with XRD diffractometer (Rigaku D/MAX-2400 (Cu K*α*)). The surface imaging of the nanocomposites was proceeded via FE-SEM (Lyra3, Tescan, Brno, Czech Republic), linked with energy-dispersive X-ray (EDX) and using a transmission electron microscope (TEM and HRTEM) (FEI Titan ST Microscopes). The magnetic measurements of the SGFO/CTTFO nanocomposites were carried out using Quantum Design PPMS DynaCool-9, coupled with a head of vibrating sample magnetometer (VSM).

## 3. Results and Discussion

### 3.1. Phase Characterization

The structure of hard/soft spinel ferrite (CNGaGdFO)*_x_*/(ZFO)*_y_* (*x:y* = 1:1, 1:2, 1:3, 2:1, 3:1 and 4:1) NCs formed by both sol-gel and PLAL methods were carried out via X-ray diffractometer as viewed in Figure 1. Each compositions exhibited single phase of spinel ferrite without detecting of any second phases. It is showed that there are a diverse in the intensity of some peaks of the of hard/soft ferrite structure in each method with changing the fraction between hard and soft. The lattice parameters, crystallite size and cell volume for both hard/soft NCs compositions of were calculated theoretical teared to X-ray experimental data through Match 3! And full-proof software as listed in Table 1. It is found that the lattice parameters NCs fabricated by sol-gel is increased for *x:y* = 1:1, 1:2 and 1:3 then decreased for *x:y* = 2:1, 3:1 and 4:1 due to redistribution of ions into the spinel lattice. However, the lattice parameters NCs produced PLAL are increased with varying the fraction between hard and soft as results of stress which expanded spinel lattice. The crystallite size was estimated via Scherrer’s equation, and it found in the range of 38 to 60 nm for samples that prepared one-pot sol-gel auto-combustion and 55 to 88 nm for PLAL. It is worth mentioning that the crystallite size of the NCs samples synthesized via PLAL are larger than the other NCs. In the PLAL technique, the plumes of plasma were produced an acoustic cavitation bubble in water medium which led to transfer the high electronic energy into lattice because of the interaction among particles. When these bubbles reached a critical volume, it will generate ultrasonic shockwaves in the water medium that helped in the formation of nanocomposites. Accordingly, the kinetic evolution of laser-induced plasma played role in the physical properties of nanocomposites for example particle size and particle shape.

### 3.2. Microstructural Features

The SEM analysis of (CNGaGdFO)*_x_*/(ZFO)*_y_* (*x:y* = 1:1, 1:2 and 2:1) NCs formed by both sol-gel and PLAL procedures as observed in Figure 2. Both group of samples showed assemblies of cubic particles with consolidated distribution. It is apparent that the samples prepared with sol-gel have a fine particles size in comparison with the PLAL. The elemental compositions of (CNGaGdFO)*_x_*/(ZFO)*_y_* (*x:y* = 1:1) NCs formed by both sol-gel and PLAL were verified utilizing EDX and elemental mapping as presented in Figure 3. It revealed the contained elements in NCs such as Co, Ni, Ga, Gd, Fe, Zn and O with no impurity. The phase and morphology of (CNGaGdFO)*_x_*/(ZFO)*_y_* (*x:y* = 1:2) NCs formed by both sol-gel and PLAL were investigated by TEM and HR-TEM as imaged in Figure 4. These outcomes supporting results from SEM and XRD.

### 3.3. Magnetic Properties

#### 3.3.1. Room-Temperature Field Dependences

Room temperature M-H curves of CNGaGdFO and ZFO SFNPs that were synthesized by sol-gel method are given in Figure 5a,b, respectively. Hysteresis curves were recorded under the experimental conditions of ±70 kOe acting dc field and 300 K temperature. S-like shape of hysteresis curve belonging to CNGaGdFO SFNPs mislead the person that this sample has superparamagnetic nature. However, enlarged view reveals the ferrimagnetic nature of this sample due to remarkable magnitude of coercivity (*H*_C_ = 625 Oe) and remnant magnetization (*M*_r_ = 17.75 emu/g). Maximum measured magnetization corresponding to 70 kOe of external field is 59.85 emu/g. Specific saturation magnetization (*M*_S_) of sample could be estimated via an additional analysis performed on hysteresis curves. Initially, *M* vs. 1/*H*^2^ plots for the high magnetic field range of [60 kOe–70 kOe] are drawn. Fitting linear lines to those plots intercept the magnetization axis for zero magnitude of 1/*H*^2^ and assign the estimate value of *M*_S_. For CNGaGdFO SFNPs, saturation magnetization was estimated as 60.63 emu/g. This value is very close to magnetization magnitude corresponding to 70 kOe and consistent with soft magnetization characteristic of NP sample. On the other hand, we observe an almost linear shaped magnetization curve belonging ZFO SFNPs. A linear line was also added to Figure 5b that enable reader to notice easily about the paramagnetic feature of soft Zn spinel ferrite at 300 K. This case can also be attributed normal spinel structure of ZFO SFNPs with Zn^2+^ in tetrahedral sites and Fe^3+^ in octahedral sites [16].

M-H hysteresis data belonging to (Co_0.5_Ni_0.5_Ga_0.01_Gd_0.01_Fe_1.98_O_4_)*_x_*/(ZnFe_2_O_4_)*_y_* NCs synthesized by sol-gel and PLAL methods are presented in Figure 6a,b, respectively. Especially enlarged views reveal the ferrimagnetic characteristics of all composite samples that were obtained by both methods.

All spectra have remarkable coercivities and remnant magnetizations. It is observed that coercivity decreases from 340 Oe to 150 Oe with increasing *y* ratio and increases from 67 Oe to 500 Oe with increasing *x* ratio for NCs that were synthesized by sol-gel method. Remnant magnetizations decrease from 20 emu/g to 8.7 emu/g with increasing *x* ratio and decrease from 14.3 emu/g to 8.2 emu/g with increasing *y* ratio for these NCs. On the other hand, *H*_C_ or *M*_r_ values for the composite samples that were obtained by PLAL method do not exhibit a direct or an indirect proportionality with increasing *x* or *y* ratios. Coercivities are in a range of [437–570] Oe and remnant magnetizations are in a range of [2.4–11.6] emu/g. It is clear that all NCs have smaller H_C_ values compared to the 625 Oe coercivity of Co_0.5_Ni_0.5_Ga_0.01_Gd_0.01_Fe_1.98_O_4_ NPs. However, (Co_0.5_Ni_0.5_Ga_0.01_Gd_0.01_Fe_1.98_O_4_)_x/_(ZnFe_2_O_4_)_y_ NC with (1:1) ratio and synthesized by sol-gel route has 20 emu/g remnant magnetization that is a larger value compared to the 17.75 emu/g of Co_0.5_Ni_0.5_Ga_0.01_Gd_0.01_Fe_1.98_O_4_ NPs.

300 K saturation magnetizations belonging all composite samples were specified by plotting *M* vs. 1/*H*^2^ graphs again. All plots including the one that belongs to Co_0.5_Ni_0.5_Ga_0.01_Gd_0.01_Fe_1.98_O_4_ NPs and fitting lines that determine the saturation magnetization magnitudes are given in Figure 7. Saturation magnetizations increase from 59.93 emu/g to 66.63 emu/g with increasing *x* ratio and decrease from 42.16 emu/g to 22.13 emu/g with increasing *y* ratio for these NCs that were obtained by sol-gel route. For the NCs that were obtained by laser ablation route, we do not observe monotonic proportionality with increasing *x* or *y* ratios, either. However, the magnitudes of *M*_S_ data are between a minimum of 19.85 emu/g belonging to sample with (3:1) ratio and a maximum of 45.80 emu/g belonging to sample with (1:3) ratio. All estimated saturation magnetization data are given in Table 2 and one can easily notice that nanocomposite samples that were synthesized by sol-gel method have larger *M*_r_ and *M*_S_ orders with respect to samples that were synthesized by laser ablation method. In general, the resultant saturation magnetization depends on many factors such as temperature, particle size, type of domain structure, fraction of disordered spins on the surface, applied processes during synthesis NCs, chemical composition, etc. Different order of crystallite/particle sizes originated by different processes and varying fraction of disordered surface spins seem to be main factors that cause to variation at magnitude of magnetizations.

Net magnetic moment is represented by a parameter that is called as magneton number (*n*_B_) and it is known that net magnetic moment of paramagnetic ZnFe_2_O_4_ NPs is 0 μ_B_ [17,18]. On the other hand, *n*_B_ of magnetically much harder Co_0.5_Ni_0.5_Ga_0.01_Gd_0.01_Fe_1.98_O_4_ NPs were calculated as equal to 2.56 μ_B_ by a well-known equation as given below [19,20,21],
(1)$$n_{B}=\left(\frac{MW\times M_{S}}{5585}\right)\left(\mu B\right)$$
in Equation (1), *MW* is molecular weight and μ_B_ is Bohr magneton as unit of magneton number. It is not ideal to use same expression to calculate a net magnetic moment for nanocomposites. In the literature, an expression depends on fraction of hard and soft ferrites is given below,
(2)$$n_{B,C}=n_{B,h}\left(1-f_{s}\right)+n_{B,s}\cdot f_{s}$$
where $n_{B,h}$ and $n_{B,s}$ are magneton numbers of hard and soft ferrites, respectively. $f_{s}$ is weight fraction (percentage) of soft ferrite phases. However, this expression does not give consistent magneton values with respect to estimated *M*_S_ values especially according to varying *x*: *y* fractions, either.

Another ratio which is obtained by dividing *M*_r_ by *M*_S_ is known as squareness ratio (SQR). This dimensionless quantity informs especially about the magnetic domain structure of investigated magnetic nanoparticles. A value of 0.5 is assigned by Stoner-Wohlfarth as critical value to exhibit a single-domain nature by the NPs [22,23]. The SQR magnitudes are between minimum of 0.160 and maximum 0.371 for the samples obtained by sol-gel route. Smaller order of SQR values were calculated in a range of [0.121–0.256] for the samples obtained by PLAL method. Hence one can claim that our nanoparticle samples (except ZnFe_2_O_4_) or all other NC samples have multi-domain structure at 300 K according to Stoner-Wohlfarth theory.

Magnetocrystalline or simply internal anisotropy field, $H_{a}$ of a magnetic material is responsible to hold the magnetic moments in the domains as aligned in particular directions and for the appearance of the coercivity in nanoparticle. Further, magnetic hardness level is a conclusion of coercivity in the magnetic material. Once the effective magnetocrystalline anisotropy constant (*K*_eff_) is specified, subsequently the magnitude of $H_{a}$ could be easily determined using the equation that correlates three magnetic parameters: *H*_C_, *M*_S_ and *K*_eff_, as given in Equation (3) [24,25]. Equation (3) is used for random particles that have the alignment of c-axis as unordered.
(3)$$H_{C}=0.64\frac{K_{\mathrm{eff}}}{M_{s}}\;\left(\mathrm{Oe}\right)$$

Hence, *K*_eff_ magnitudes were obtained for Co_0.5_Ni_0.5_Ga_0.01_Gd_0.01_Fe_1.98_O_4_ sample and all other nanocomposites by putting estimated *M*_S_ and measured *H*_C_ data in Equation (3). Co_0.5_Ni_0.5_Ga_0.01_Gd_0.01_Fe_1.98_O_4_ has largest coercivity and largest corresponding *K*_eff_ value of 5.92 × 10^4^ Erg/g. All nanocomposite samples that were synthesized by both sol-gel and laser methods have small magnitudes of *K*_eff_ at the order 10^4^ Erg/g. $H_{a}$ can be evaluated by using another well-known expression [26,27,28],
(4)$$H_{a}=\;\frac{2K_{\mathrm{eff}}}{M_{s}}\;\;\left(\mathrm{Oe}\right)$$

Among the samples obtained by sol-gel route, $H_{a,\max}$ with a magnitude of 1953 Oe belongs to sample Co_0.5_Ni_0.5_Ga_0.01_Gd_0.01_Fe_1.98_O_4_ NPs as expected and $H_{a,\min}$ with a magnitude of 210 Oe belongs to sample with a ratio of (1:3). However, the composite samples obtained by PLAL method have internal anisotropy fields in a narrow range between 1366 Oe and 1781 Oe. In general, both type samples exhibit nonmonotonic correlation with respect to x:y fraction. All derived $H_{a}$, *K*_eff_ and SQR 300 K data are included in the Table 2 with other magnetic parameters.

#### 3.3.2. Low-Temperature Field Dependences

Low temperature M-H curves belonging to Co_0.5_Ni_0.5_Ga_0.01_Gd_0.01_Fe_1.98_O_4_ and ZnFe_2_O_4_ NPs that were synthesized by sol-gel method are given in Figure 8a,b, respectively.

Hysteresis curves were recorded for the acting external dc field of ±70 kOe and at 10 K. Co_0.5_Ni_0.5_Ga_0.01_Gd_0.01_Fe_1.98_O_4_ NPs exhibit ferrimagnetic phase at present measurement conditions. Enlarged view of Co_0.5_Ni_0.5_Ga_0.01_Gd_0.01_Fe_1.98_O_4_ NP facilitates the determination of significant magnitude of coercivity as 2207 Oe and remnant magnetization as 34.54 emu/g. Maximum measured magnetization corresponding to 70 kOe of field is 67.32 emu/g and estimated *M*_S_ value is 69.84 emu/g with corresponding magneton number of 2.95 μ_B_ at 10 K. SQR magnitude was calculated as 0.495 by assigning the almost single-domain nature for nanoparticle sample. Co_0.5_Ni_0.5_Ga_0.01_Gd_0.01_Fe_1.98_O_4_ has also the largest *K*_eff_ value of 2.4 × 10^5^ Erg/g and third large internal anisotropy field of 6897 Oe among all samples that were synthesized by sol-gel approach.

*M*_S_ value of ZnFe_2_O_4_ NPs was estimated as 42.50 emu/g with a corresponding *n*_B,min_ of 1.83 μ_B_ among all samples that were obtained by sol-gel route. S-like shape of hysteresis curve belonging to ZnFe_2_O_4_ reveals the superparamagnetic (SPM) nature of this soft spinel NPs. In SPM phase, ferrite nanoparticle has a single domain structure. The order of exchange interactions between magnetic moments is not enough for nanoparticle to exhibit ferrimagnetic behavior. In addition, the presence or absence of interparticle interactions urge different magnetic phases such as a conventional superparamagnetic behavior or spin glass-like phase.

10 K M-H hysteresis curves belonging to all composite samples that were synthesized by both sol-gel and PLAL methods and their enlarged views are given in Figure 9a,b, respectively. Expectedly, NCs perpetuate their ferrimagnetic characteristics at 10 K. Varying *H*_C_ or *M*_r_ values for the composite samples that were obtained by sol-gel method do not exhibit exactly same observed behavior of 300 K data. Coercivities are in a range of [571–3564] Oe and remnant magnetizations are in a range of [14.55–42.27] emu/g. Saturation magnetizations are in a range of [30.59–118.71] emu/g, Figure 10a. However, NCs that were obtained by PLAL exhibit a nonmonotonic proportionality with respect to increasing *x* or *y* ratios at 10 K again. Coercivities are in a range of [485–1785] Oe and remnant magnetizations are in a range of [5.5–23.5] emu/g. Saturation magnetizations are in a range of [44.59–64.55] emu/g, Figure 10b. Hence, one can say that 10 K magnetic parameters are much larger with respect to already determined 300 K data due to absence of demagnetizing effect of heat at high temperatures. In addition, NCs that were synthesized by sol-gel method have larger magnetic parameters than the NCs that were synthesized by PLAL method at 10 K. Dimensionless SQR data are in the ranges of [0.356–0.557] and [0.104–0.365] reflecting the deviation from single-domain nature for NCs obtained by sol-gel and PLAL methods, respectively. *K*_eff_ values vary in a narrow range between 1.03 × 10^5^ Erg/g and 2.02 × 10^5^ Erg/g while $H_{a}$ values vary in a large range between 1784 Oe and 11,138 Oe for the NCs obtained by sol-gel route. As last comparison, *K*_eff_ values for the NCs obtained by PLAL route have a range between 0.43 × 10^5^ Erg/g and 1.79 × 10^5^ Erg/g while $H_{a}$ values vary between 1516 Oe and 5578 Oe. One can say that NCs synthesized by sol-gel approach are magnetically harder than the NCs synthesized by PLAL approach. All measured or determined 10 K magnetic parameters are given in Table 3.

#### 3.3.3. Temperature Dependences

Temperature dependent zero field cooling (ZFC) and field cooling (FC) measurements can specify different magnetic phases that are exhibited by the same magnetic host at different temperature conditions. The recorded ZFC and FC spectra belonging to Co_0.5_Ni_0.5_Ga_0.01_Gd_0.01_Fe_1.98_O_4_ and ZnFe_2_O_4_ NPs that were synthesized by sol-gel method are given in Figure 11a,b, respectively. When ZFC measurements are executed, initially magnetic nanoparticle is cooled in zero magnetic field until 10 K as minimum temperature for our investigations and subsequently the magnetization data are registered by increasing temperature until a maximum of 325 K for our investigations while a 100 Oe dc field exposes to nanoparticles, simultaneously. After recording of ZFC data, NPs and NCs are cooled at 100 Oe of external dc field until 10 K and then FC curves were recorded with increasing temperature until 325 K and the acting field was kept constant at 100 Oe for our investigations. The maximum magnetization point reached by ZFC spectra is assigned as blocking temperature (*T*_B_) for magnetic nanoparticles. At *T*_B_ the thermal energy (*k*_B_T) equals to energy barrier for magnetic moment reversal associated with the total crystal anisotropy energy (*E*_B_ = *K*_eff_V = *k*_B_T). If measurement temperature is above *T*_B_, then NPs exhibit SPM phase consistent with the Curie-Weiss law [29]. In SPM phase, the remnant magnetizations or coercivities are not detected. Magnetic nanoparticles can exhibit their ferro-ferrimagnetic phases at temperature ranges below *T*_B_ [30].

In Figure 11a, a splitting and a large irreversibility between ZFC and FC curves is easily noticed. FC magnetization decreases until 200 K and then stays almost same. Further, one cannot see a summit point at ZFC curve assigning a blocking temperature value for Co_0.5_Ni_0.5_Ga_0.01_Gd_0.01_Fe_1.98_O_4_ NPs. Instead, ZFC magnetization decreases almost linearly until 10 K. The dispersion at ZFC and FC magnetizations specifies a poly-disperse character for magnetic NPs with a related distribution in particle dimension and individual anisotropy axes [31]. In addition, this case is consistent with ferrimagnetic property of this nanoparticle sample even at 300 K. However, a peak value is observed on ZFC magnetization curve at 19.23 K which is called as spin freezing temperature (*T*_f_) for ZnFe_2_O_4_ NPs that were synthesized by sol-gel method. M_ZFC_(T) and M_FC_(T) curves diverge around 65 K before the temperature drops to *T*_f_ = 19.23 K. A sharp cusp at low temperatures is observed for the sample from the temperature dependence of the low field dc magnetization. Such a FC-ZFC difference at low temperatures is attributed to spin glass or spin glass-like phases in the magnetic material [32].

ZFC and FC measurements were also conducted for NCs synthesized by both sol-gel and PLAL methods and registered spectra are presented in Figure 12a,b, respectively. Significant dispersion at ZFC and FC magnetizations are also observed almost at every curve. Dispersion magnitudes belonging to composite samples with 1:1 and 1:3 ratios seem to be larger with respect to other NPs. None of spectra includes a T_f_ value for the samples that were synthesized by sol-gel method. Among the samples that were synthesized by PLAL route, one can talk about the T_f_ temperatures of 95.0 K, 94.7 K, 65.6 K and 68.0 K correspond *x*:*y* ratios of 1:1, 1:2, 2:1 and 3:1, respectively. When temperature decreases below T_f_, the antiferromagnetic interactions become stronger in octahedral B-sites of spinel ferrite. This case causes to a collective spin freezing and remarkable decrease at magnetization. However, FC magnetization spectra are expected to keep their increasing trend and a maximum magnetization should be recorded at minimum cooling temperature. We observe this case at FC magnetization spectra recorded from samples that were synthesized by PLAL method.

## 4. Conclusions

The spinel ferrite nanocomposites (CNGaGdFO)*_x_*/(ZFO)*_y_* (*x*:*y* = 1:1, 1:2, 1:3, 2:1, 3:1 and 4:1) and (ZFO) SFNPs, (CNGaGdFO) SFNPs individually were produced with sol-gel and Green pulsed laser ablation in liquid (PLAL) methodologies. The constructed phases and morphology were analyzed and investigated in detail via XRD, SEM, TEM and HR-TEM. Soft ZnFe_2_O_4_ ferrite nanoparticle sample exhibits paramagnetic phase at 300 K and SPM phase with single domain nature at low temperatures. Other magnetically harder Co_0.5_Ni_0.5_Ga_0.01_Gd_0.01_Fe_1.98_O_4_ NPs exhibits ferrimagnetic phases at 300 K with multi-domain and at 10 K with single-domain nature. At most M-T spectra a significant dispersion between ZFC and FC magnetizations are observed. This case can be attributed to poly-disperse characteristics of magnetic NPs and NCs with a correlated distribution in nanoparticle dimension and individual anisotropy axes. Magnetic parameters belonging to NCs that were synthesized by sol-gel method are usually stronger than magnetic parameters of the NCs that were synthesized by PLAL method at both temperatures. On the other hand, 300 K coercivities are relatively larger for some NCs that were synthesized by PLAL method. Some nanocomposite samples have larger *M*_S_, *M*_r_ and *H*_C_ values as compared to magnetic data of Co_0.5_Ni_0.5_Ga_0.01_Gd_0.01_Fe_1.98_O_4_ NPs. According 10 K data, the samples that were obtained by sol-gel route and having the *x:y* ratios of 3:1 and 4:1 are magnetically harder than Co_0.5_Ni_0.5_Ga_0.01_Gd_0.01_Fe_1.98_O_4_ NPs. In any case, one can classify all NP or NC samples as magnetically soft material. This kind of materials can be used as functional magnetic materials for target drug delivery, hyperthermia effect and for microwave applications.

## Figures and Tables

**Figure 1 nanomaterials-11-02461-f001:**
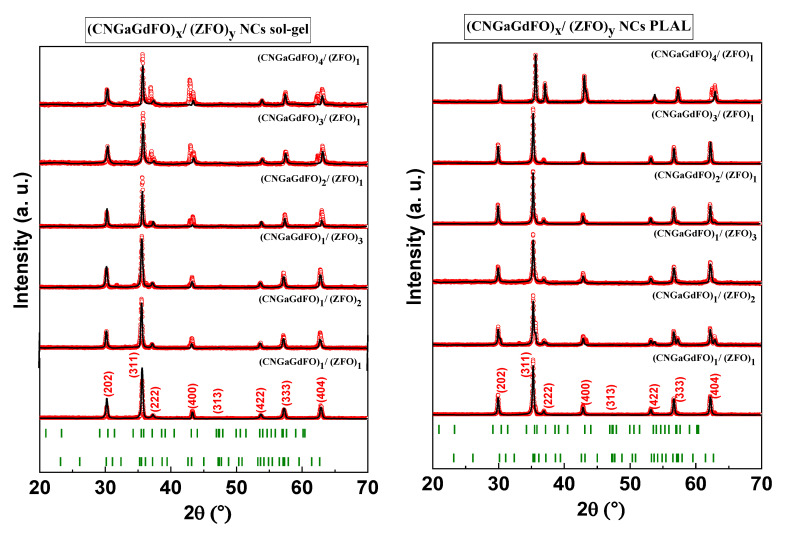
XRD of (CNGaGdFO)*_x_*/(ZFO)*_y_* (*x:y* = 1:1, 1:2, 1:3, 2:1, 3:1 and 4:1) NCs prepared by sol-gel and PLAL.

**Figure 2 nanomaterials-11-02461-f002:**
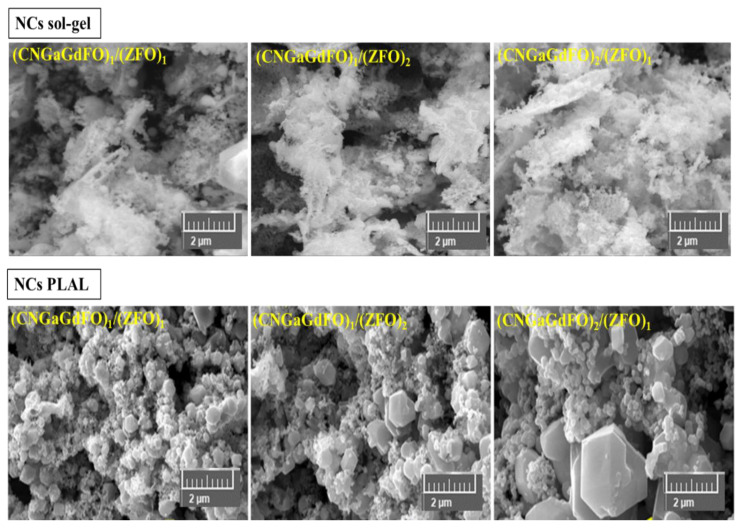
SEM of (CNGaGdFO)*_x_*/(ZFO)*_y_* (*x:y* = 1:1, 1:2 and 2:1) NCs prepared by sol-gel and PLAL.

**Figure 3 nanomaterials-11-02461-f003:**
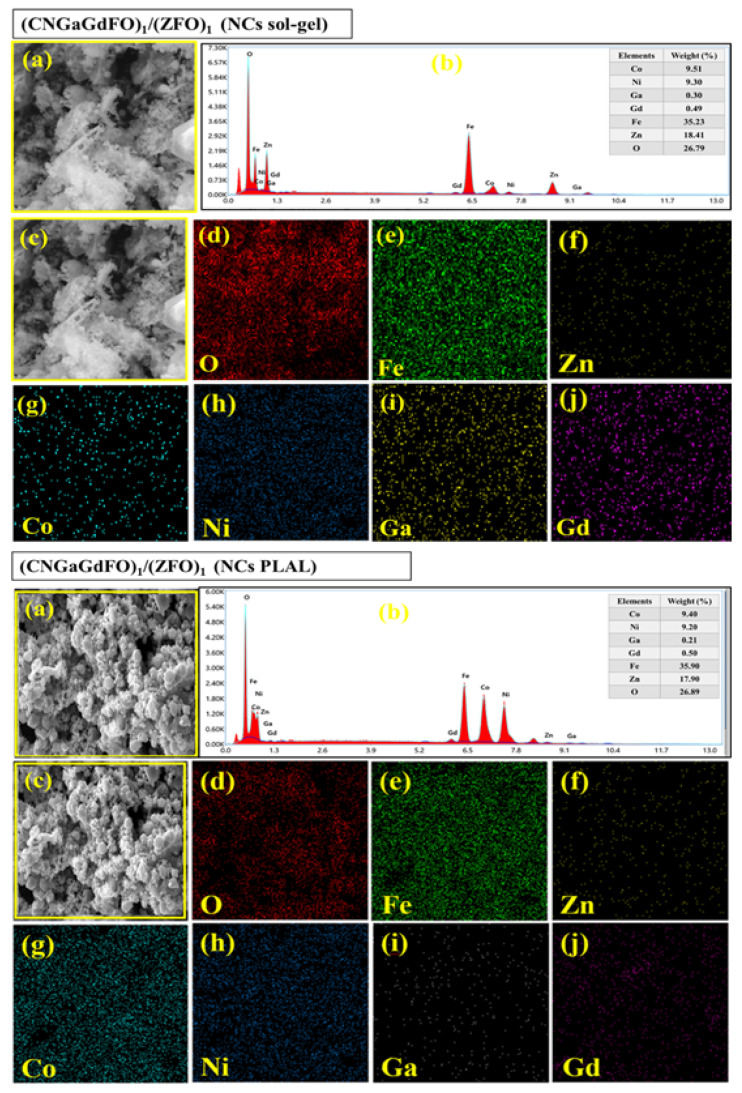
EDX and elemental mapping (EM) of (CNGaGdFO)*_x_*/(ZFO)*_y_* (*x:y* = 1:1) NCs prepared by sol-gel (top) and PLAL (down): (**a**) general SEM image; (**b**) EDX spectra; (**c**) detailed SEM image for EM; (**d**) oxygen distribution on EM; (**e**) Fe distribution on EM; (**f**) Zn distribution on EM; (**g**) Co distribution on EM; (**h**) Ni distribution on EM; (**i**) Ga distribution on EM; (**j**) Gd distribution on EM.

**Figure 4 nanomaterials-11-02461-f004:**
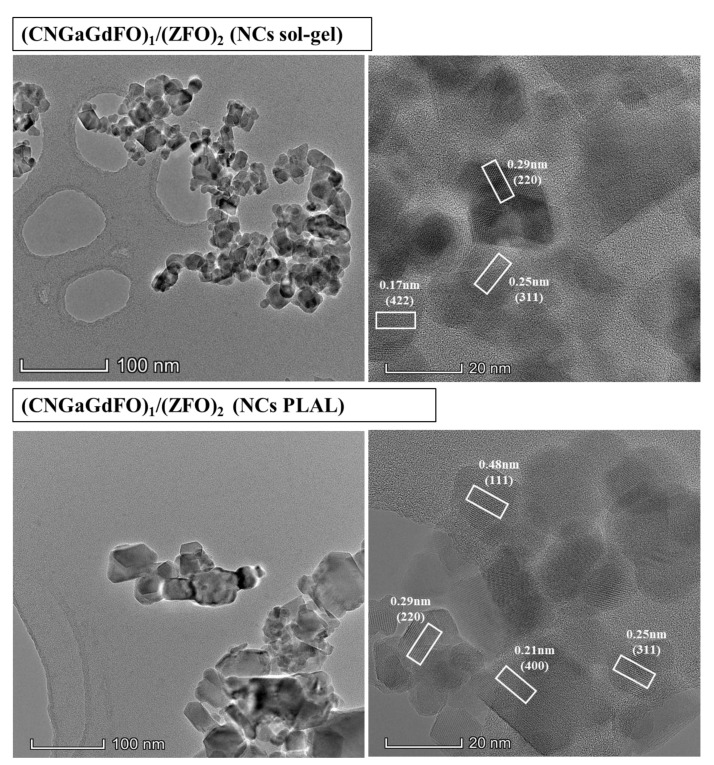
TEM and HR-TEM of (CNGaGdFO)*_x_*/(ZFO)*_y_* (*x:y* = 1:2) NCs prepared by sol-gel and PLAL.

**Figure 5 nanomaterials-11-02461-f005:**
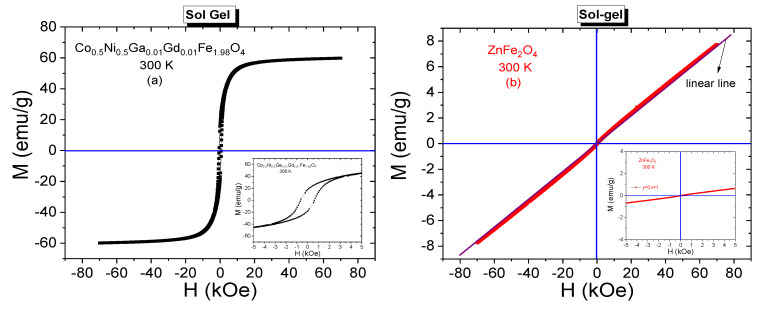
M-H curves recorded at 300 K from (**a**) CNGaGdFO SFNPs and (**b**) ZFO SFNPs synthesized by sol-gel method. Enlarged views were embedded, too.

**Figure 6 nanomaterials-11-02461-f006:**
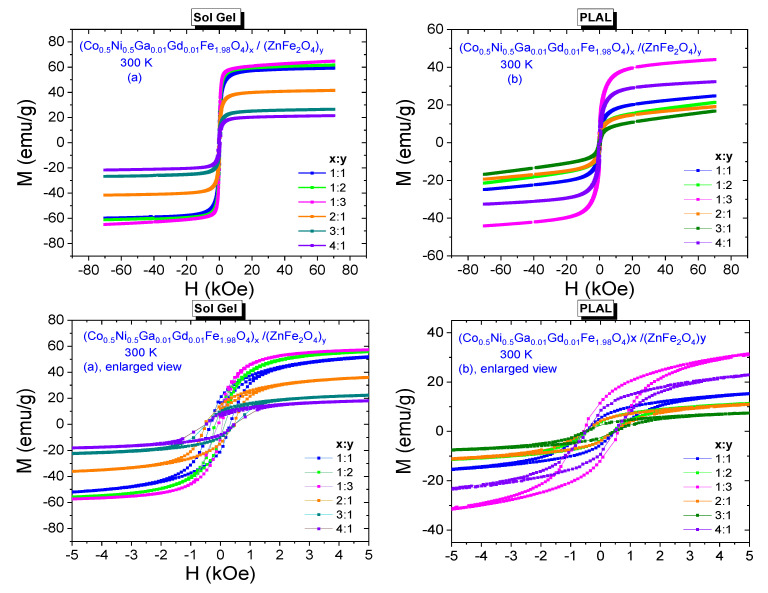
Room temperature (T = 300 K) M-H curves recorded from (Co_0.5_Ni_0.5_Ga_0.01_Gd_0.01_Fe_1.98_O_4_)*_x_*/(ZnFe_2_O_4_)*_y_* NCs synthesized by (**a**) sol-gel method (**b**) PLAL method, including enlarged views under them.

**Figure 7 nanomaterials-11-02461-f007:**
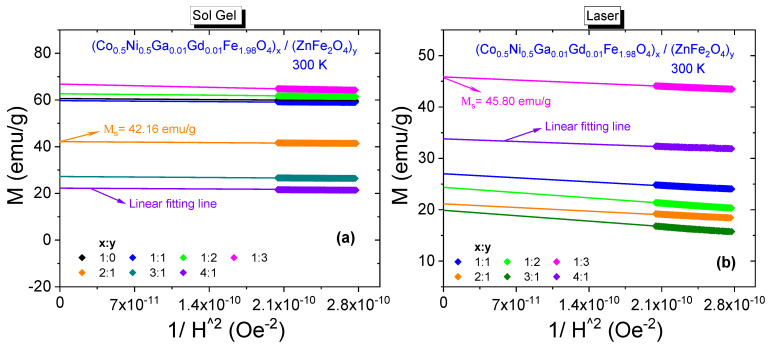
*M* vs. 1/*H*^2^ plots for (Co_0.5_Ni_0.5_Ga_0.01_Gd_0.01_Fe_1.98_O_4_)*_x_*/(ZnFe_2_O_4_)*_y_* NCs synthesized by (**a**) sol-gel method and (**b**) PLAL method, 300 K.

**Figure 8 nanomaterials-11-02461-f008:**
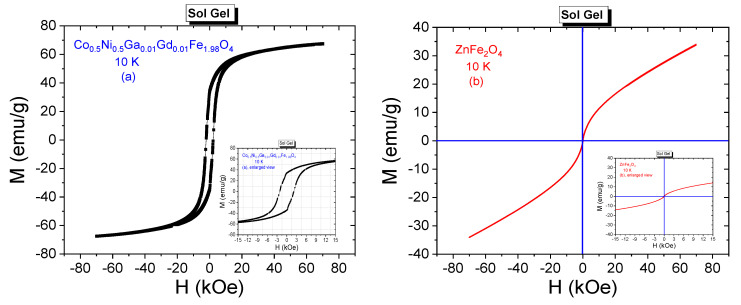
M-H curves recorded at 10 K from (**a**) Co_0.5_Ni_0.5_Ga_0.01_Gd_0.01_Fe_1.98_O_4_ and (**b**) ZnFe_2_O_4_ NPs synthesized by sol-gel method. Enlarged views were embedded.

**Figure 9 nanomaterials-11-02461-f009:**
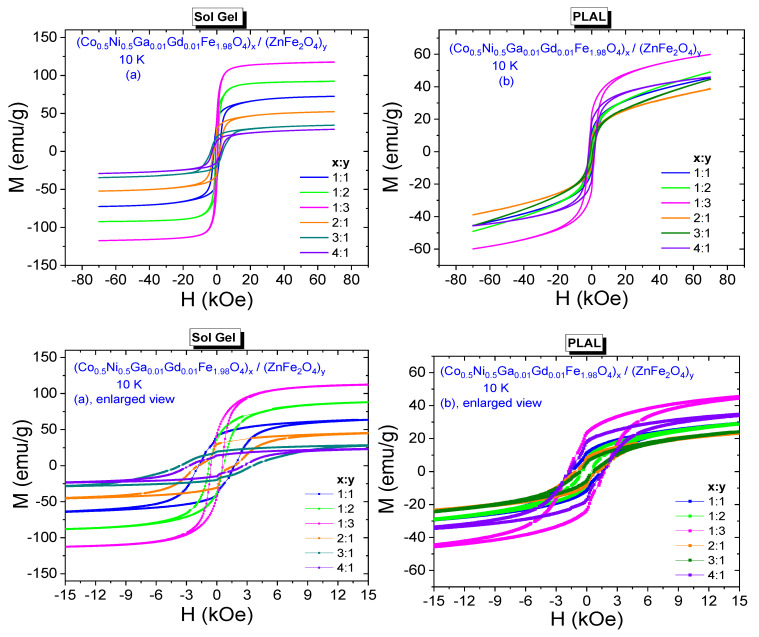
10 K M-H curves recorded from (Co_0.5_Ni_0.5_Ga_0.01_Gd_0.01_Fe_1.98_O_4_)*_x_*/(ZnFe_2_O_4_)*_y_* NCs synthesized by (**a**) sol-gel method and enlarged view (**b**) laser ablation method and enlarged view.

**Figure 10 nanomaterials-11-02461-f010:**
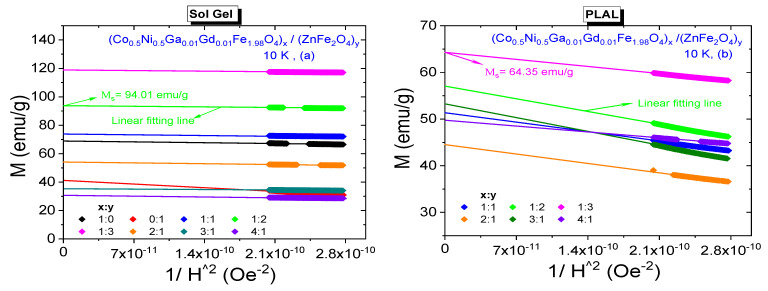
*M* vs. 1/*H*^2^ plots for (Co_0.5_Ni_0.5_Ga_0.01_Gd_0.01_Fe_1.98_O_4_)*_x_*/ZnFe_2_O_4_)*_y_* NCs synthesized by (**a**) sol-gel method and (**b**) PLAL method, 10 K.

**Figure 11 nanomaterials-11-02461-f011:**
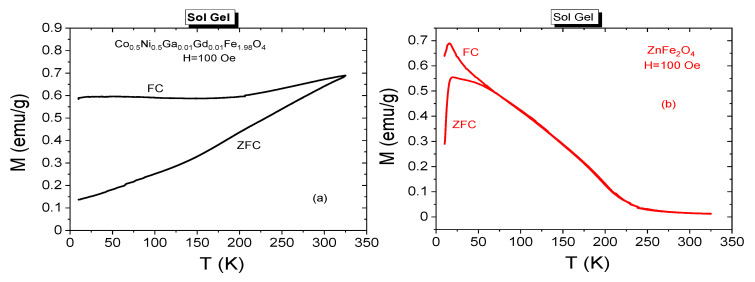
Temperature dependence of ZFC and FC magnetizations for (**a**) Co_0.5_Ni_0.5_Ga_0.01_Gd_0.01_Fe_1.98_O_4_ and (**b**) ZnFe_2_O_4_ NPs synthesized by sol-gel method.

**Figure 12 nanomaterials-11-02461-f012:**
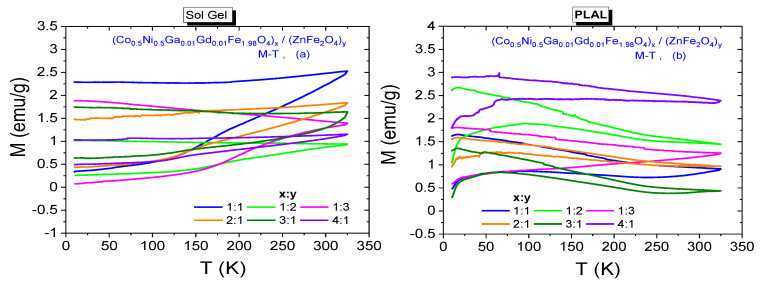
Temperature dependence of ZFC and FC magnetizations for (Co_0.5_Ni_0.5_Ga_0.01_Gd_0.01_Fe_1.98_O_4_)*_x_*/(ZnFe_2_O_4_)*_y_* NCs synthesized by (**a**) sol-gel method and (**b**) laser ablation method, respectively.

**Table 1 nanomaterials-11-02461-t001:** Refined structure parameters of CNGaGdFeO NPs, ZFO NPs and (CNGaGdFO)*_x_*/(ZFO)*_y_* (*x:y* = 1:1, 1:2, 1:3, 2:1, 3:1 and 4:1) NCs prepared by sol-gel and PLAL.

Product	Sol–Gel			PLAL		
	D_XRD_ (nm)	a (Å)	V (Å^3^)	D_XRD_ (nm)	a (Å)	V(Å^3^)
**(CNGaGdFeO)_1_/(ZFO)_1_ NCs**	38.9	8.3603	584.3400	70.1	8.4366	600.4810
**(CNGaGdFeO)_1_/(ZFO)_2_ NCs**	48.4	8.3628	585.7145	75.9	8.4375	600.6861
**(CNGaGdFeO)_1_/(ZFO)_3_ NCs**	42.1	8.3667	585.6830	88.6	8.4379	600.7714
**(CNGaGdFeO)_2_/(ZFO)_1_ NCs**	59.1	8.3470	581.5556	86.3	8.4423	601.6946
**(CNGaGdFeO)_3_/(ZFO)_1_ NCs**	59.9	8.3406	580.2148	85.2	8.4585	605.1651
**(CNGaGdFeO)_4_/(ZFO)_1_ NCs**	52.4	8.3400	576.7385	55.8	8.4654	683.3110

**Table 2 nanomaterials-11-02461-t002:** Room temperature magnetic parameters (Co_0.5_Ni_0.5_Ga_0.01_Gd_0.01_Fe_1.98_O_4_)*_x_*/(ZnFe_2_O_4_)*_y_* NCs synthesized by both sol-gel and PLAL.

** *x:y* **	**MW** **g/mol**	**Sol-Gel**					
		**M_r_** **emu/g**	**H_C_** **Oe**	**M_S_** **emu/g**	**SQR** ***M*_r_/*M*_s_**	***K*_eff_ × 10^4^** **Erg/g**	** *H* _a_ ** **Oe**
**1:0**	235.65	17.75	625	60.63	0.294	5.92	1953
**1:1**	476.73	20.0	340	59.93	0.334	3.18	1062
**1:2**	717.81	10.0	150	62.69	0.160	1.47	469
**1:3**	958.89	8.7	67	66.53	0.131	0.7	210
**2:1**	712.39	14.3	472	42.16	0.339	3.11	1475
**3:1**	948.04	10.0	480	27.19	0.368	2.04	1500
**4:1**	1183.7	8.2	500	22.13	0.371	1.73	1562
** *x:y* **	**MW** **g/mol**	**PLAL**					
		** *M* _r_ ** **emu/g**	** *H* _C_ ** **Oe**	** *M* _S_ ** **emu/g**	**SQR** **M_r_/M_s_**	***K*_eff_ × 10^4^** **Erg/g**	** *H* _a_ ** **Oe**
**1:1**	476.73	5.4	531	26.97	0.200	2.24	1659
**1:2**	717.81	3.5	437	24.29	0.144	1.66	1366
**1:3**	958.89	11.6	570	45.80	0.253	4.08	1781
**2:1**	712.39	3.9	492	21.32	0.183	1.64	1537
**3:1**	948.04	2.4	467	19.85	0.121	1.45	1459
**4:1**	1183.7	8.6	480	33.61	0.256	2.52	1500

**Table 3 nanomaterials-11-02461-t003:** 10 K magnetic parameters (Co_0.5_Ni_0.5_Ga_0.01_Gd_0.01_Fe_1.98_O_4_)*_x_*/(ZnFe_2_O_4_)*_y_* NPs synthesized by both sol-gel and PLAL.

** *x:y* **	**MW** **g/mol**	**Sol-Gel**					
		** *M* _r_ ** **emu/g**	** *H* _C_ ** **Oe**	** *M* _S_ ** **emu/g**	**SQR** **M_r_/M_s_**	***K*_eff_ × 10^5^** **Erg/g**	** *H* _a_ ** **Oe**
1:0	235.65	34.54	2207	69.84	0.495	2.40	6897
0:1	241.08	-	-	42.50	-	-	-
1:1	476.73	41.15	1750	73.87	0.557	2.02	5469
1:2	717.81	36.75	702	94.01	0.391	1.03	2194
1:3	958.89	42.27	571	118.7	0.356	1.06	1784
2:1	712.39	30.00	1709	54.06	0.555	1.44	5341
3:1	948.04	19.90	3564	35.78	0.556	1.99	11138
4:1	1183.7	14.55	2663	30.59	0.476	1.27	8321
** *x:y* **	**MW** **g/mol**	**PLAL**					
		** *M* _r_ ** **emu/g**	** *H* _C_ ** **Oe**	** *M* _S_ ** **emu/g**	**SQR** ***M*_r_/*M*_s_**	***K*_eff_ × 10^5^** **Erg/g**	** *H* _a_ ** **Oe**
1:1	476.73	11.2	1490	51.43	0.218	1.20	4656
1:2	717.81	7.8	485	57.10	0.137	0.43	1516
1:3	958.89	23.5	1785	64.35	0.365	1.79	5578
2:1	712.39	8.0	1070	44.59	0.179	0.74	3344
3:1	948.04	5.5	685	53.04	0.104	0.57	2140
4:1	1183.7	18.0	1682	49.62	0.363	1.30	5256

## Data Availability

Not applicable. Data could be shared upon reasonable request.

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
