# Peer review of "Structural and Magnetic Properties of Co0.5Ni0.5Ga0.01Gd0.01Fe1.98O4/ZnFe2O4 Spinel Ferrite Nanocomposites: Comparative Study between Sol-Gel and Pulsed Laser Ablation in Liquid Approaches"

_nanomaterials, 2021, doi:10.3390/nano11092461_

Round 1

Reviewer 1 Report

The manuscript  Structural and magnetic properties of Co0.5Ni0.5Ga0.01Gd0.01Fe1.98O4/ZnFe2O4 spinel ferrite  nanocomposites: Comparative study between sol-gel and Pulsed Laser Ablation in Liquid approaches. Moreover, methods and experiments are documented in detail. Results are presented in an appropriate manner, clearly explaining the results of the study. The manuscript is written in a direct and active style. However, some corrections will be needed, see recommendations below. Overall, the paper offers enough details of its methodology to reproduce the experiments and is written comprehensively enough to be understandable. I suggest accepting this manuscript after the revision, and the authors should consider the suggestions described below:

1; Compare the traditional methods of analysis with the methods based on this subject. Maybe a table could help

2: Authors should carefully revised and corrected all the grammatical issues and follow the scientific norms in the whole manuscript

3: Conclusion should be rewrite. Please added future perspective before Conclusion

4:            Abstract: Authors need to present the key objectives of present Chapter this paper and brief summary of conclusions in abstract. (in the abstract ending, some  findings and barriers which can be tackled to improve magnetic nanocomposite were mentioned.) 

5:            Introduction: Authors need to more describe the problem statement and objectives of the paper in the last paragraph of the Introduction section. In addition, it is recommended to include a paragraph Comparative study between sol-gel literature spinel ferrite  nanocomposites in introduction.

6:            References: It is recommended to replace the old references with the latest literature.

7: Do you mains about economic process?

8 : Some of introduction on magnetic applications and … are need using from below papers, so, below papers are add to manuscript

Author Response

Referee #1 Comments and Suggestions for Authors

The manuscript  Structural and magnetic properties of Co0.5Ni0.5Ga0.01Gd0.01Fe1.98O4/ZnFe2O4 spinel ferrite  nanocomposites: Comparative study between sol-gel and Pulsed Laser Ablation in Liquid approaches. Moreover, methods and experiments are documented in detail. Results are presented in an appropriate manner, clearly explaining the results of the study. The manuscript is written in a direct and active style. However, some corrections will be needed, see recommendations below. Overall, the paper offers enough details of its methodology to reproduce the experiments and is written comprehensively enough to be understandable. I suggest accepting this manuscript after the revision, and the authors should consider the suggestions described below:

Comment 1: Compare the traditional methods of analysis with the methods based on this subject. Maybe a table could help

Response: Dear Reviewer, thank you for your comment. It was done as requested. Please see in revised version.

Comment 2:  Authors should carefully revised and corrected all the grammatical issues and follow the scientific norms in the whole manuscript

Response: Dear Reviewer, thank you for your comment. It was revised as requested. Please see in revised version.

Comment 3:  Conclusion should be rewrite. Please added future perspective before Conclusion

Response: Dear Reviewer, thank you for your comment. It was done as requested. Please see in revised version.

Comment 4:  Abstract: Authors need to present the key objectives of present Chapter this paper and brief summary of conclusions in abstract. (in the abstract ending, some  findings and barriers which can be tackled to improve magnetic nanocomposite were mentioned.) 

Response: Dear Reviewer, thank you for your comment. It was done as requested. Please see in revised version. Main object and brief summary of findings were added to the abstract part of manuscript according to remark of reviewer.

Comment 5: Introduction: Authors need to more describe the problem statement and objectives of the paper in the last paragraph of the Introduction section. In addition, it is recommended to include a paragraph Comparative study between sol-gel literature spinel ferrite  nanocomposites in introduction.

Response: Dear Reviewer, thank you for your comment. It was done as requested. Please see in revised version.

Comment 6: References: It is recommended to replace the old references with the latest literature.

Response: Dear Reviewer, thank you for your comment. It was done as requested. Please see in revised version.

Comment 7:  Do you mains about economic process?

Response: Dear Reviewer, thank you for your comment. Sorry us. But in this study we focused only on correlation between physical phenomena and didn’t think about the economic process. Hope on your understanding and support.

Comment 8:  Some of introduction on magnetic applications and … are need using from below papers, so, below papers are add to manuscript

Response: Dear Reviewer, thank you for your comment. It was done as requested. Please see in revised version.

General comment from authors:

Many thanks for your comments. All comments were useful for us and we hope it helped us do our manuscript better.

Reviewer 2 Report

The authors studied very well on molecular structure, microstructure, and  magnetism of the samples. However, ESR study is not exist in this paper. In order to investigate the origin or variation of the electronic configuration or the spin system or spin state for the samples, the ESR study is needed for deep discussion on magnetism of the samples. 

Author Response

Referee #2 Comments and Suggestions for Authors

Comment: The authors studied very well on molecular structure, microstructure, and  magnetism of the samples. However, ESR study is not exist in this paper. In order to investigate the origin or variation of the electronic configuration or the spin system or spin state for the samples, the ESR study is needed for deep discussion on magnetism of the samples. 

Response: Dear Reviewer, thank you for your comment. You are right that ESR is very important technique for magnetic properties analysis. But in previous study we do not use ESR. We focused only on microscopic magnetic analysis using VSM at wide ranges of the magnetic field and temperatures. We observed how two different magnetic phases interact to each other. The analysis of the ESR spectra for two phase composite is too complex task. Thus in this paper we demonstrate only VSM-results. Probably in further investigations we will do ESR experiments and will share by this. Hope on your understanding and support.

General comment from authors:

Many thanks for your comments. All comments were useful for us and we hope it helped us do our manuscript better.

Reviewer 3 Report

  1. The author is expected to analyze the SEM images in more detail, rather than just explain the content of the images.
  2. As for the analysis of magnetic properties, some sub-headings need to be added to the manuscript to make it more logical.
  3. Add some references where needed to make the manuscript more persuasive.
  4. Regarding magnetic properties, manuscripts lack sufficient analytical content (e.g., to analyze phenomena and explain why phenomena arise).

Author Response

Referee #3 Comments and Suggestions for Authors

Comment:1. The author is expected to analyze the SEM images in more detail, rather than just explain the content of the images.

Response: Dear Reviewer, thank you for your comment. We added some description of the SEM images as requested.

Comment: 2. As for the analysis of magnetic properties, some sub-headings need to be added to the manuscript to make it more logical.

Response: Dear Reviewer, thank you for your comment. We added sub-headings (please see 3.3.1-3.3.3).

Comment: 3. Add some references where needed to make the manuscript more persuasive.

Response: Dear Reviewer, thank you for your comment. We added some relevant references.

Comment: 4. Regarding magnetic properties, manuscripts lack sufficient analytical content (e.g., to analyze phenomena and explain why phenomena arise).

Response: Dear Reviewer, thank you for your comment. Try to analysed some reason of magnetic phenomena in this study. Please be so kind and let us share by the result with the readers. Hope on your understanding and support.

General comment from authors:

Many thanks for your comments. All comments were useful for us and we hope it helped us do our manuscript better.

Round 2

Reviewer 1 Report

I'm Please inform you that about this s paper.I have reviewer this paper carefully.I suggest this manuscript accept.

Best Regards

Reviewer 3 Report

Accept.